# The Impact of Public Health Education on Migrant Workers’ Medical Service Utilization

**DOI:** 10.3390/ijerph192315879

**Published:** 2022-11-29

**Authors:** Deshui Zhou, Lanyan Cheng, Hainan Wu

**Affiliations:** School of Finance and Public Administration, Anhui University of Finance & Economics, Bengbu 233030, China

**Keywords:** public health education, medical service utilization, migrant worker

## Abstract

Based on the dynamic monitoring survey data of China’s migrant population (CMDS) in 2017, this study analyzes the impact of public health education on migrant workers’ medical service utilization. The study found that public health education can significantly promote the utilization of migrant workers’ medical services and has a greater effect on the older generation groups, those who received secondary and higher education, and those working in first-tier cities. By distinguishing different types of public health education, it is found that smoking control education has the most obvious effect. Further differentiating disease types, the study found that the promotion effect of receiving occupational disease education is the highest, while the effect of receiving STD/AIDS education is relatively low. The mechanism test indicates that public health education has significantly improved migrant workers’ utilization of medical services by influencing their health literacy, social network, and psychological integration.

## 1. Introduction

### 1.1. Background

After the reform and opening up, the migrant workers have become a new type of labor force and made great contributions to the process of urbanization and industrialization in China. By the end of 2021, the number of migrant workers has reached 290 million, accounting for about 40% of the urban and rural employment population, which provides a solid guarantee for China’s modernization. However, due to the large number and high instability of migrant workers, there are still problems, such as inadequate protection of individual rights and interests and lack of standard public services, which makes this special group show the characteristics of “economic acceptance and social exclusion” [1]. In January 2022, the Fourteenth Five-Year Health Standardization Work Plan issued by the National Health Commission of China clearly stated that “by 2025, we should basically build a health standard system with Chinese characteristics that strongly supports the construction of a healthy China”.

As a key component of the Healthy China strategy, the supply of public health education services has become an important consideration for national health policy deployment [2]. At the beginning of 2020, the outbreak of COVID-19 had a great impact on China’s health industry, and public health education has caught much more of academia’s attention [3,4]. Public health education can improve people’s health literacy and health concerns by affecting individual and group health prevention so as to reduce disease transmission [5]. 74.50% of Chinese citizens support the promotion of public health education and training after the outbreak of COVID-19, which pushes the development of public health education to a certain extent [4].

While public health education has gained attention, the utilization of medical services is also an important part of the Healthy China strategy [6], which refers to people’s active or passive medical behavior on health problems. A low utilization rate of medical services will lead to the deterioration of individual health status and increase health inequality [7]. Scholars explored the utilization of medical services for migrant workers from different perspectives, such as medical system reform, medical insurance coverage, and population aging [8,9,10], but few discussed the relationship between public health education and migrant workers’ medical behavior. Then, can public health education affect the utilization of migrant workers’ medical services? Does the effect have any heterogeneity in different groups? What are the differences between various types of public health education in the use of migrant workers’ medical services? What are the mechanisms of public health education on migrant workers’ medical service utilization? In order to clarify the above issues, this study uses the dynamic monitoring survey data of the migrant population (CMDS) in 2017, adopts instrumental variables to solve the endogeneity, and explores the impact of public health education on migrant workers’ medical service utilization, that may deliver certain values for developing public health theories and improving relevant public health policies.

### 1.2. Literature Review

At present, there is a large gap between the quality of life of China’s migrant population and the national average living standard [11]. The level of health education of the migrant population is generally low [12], and their health status is considered worrying. Moreover, the acceptance rate of health education among China’s urban migrant population is higher than that of the rural migrant population [13]. Such a situation has called for academia’s attention to public health education. Based on the mortality data of Spain during the COVID-19 period, Luque [14] found that standardized health education can improve the ability to respond to public emergencies and reduce epidemic mortality. Zhao and Wang [15] explored the marginal utility of promoting health education to improve individual health is more significant for young people, those without higher education, and in the central and western regions. Greenberg [16] proposed that integrating social-emotional learning (SEL) with other services can significantly increase community health.

Meanwhile, although the quantity of health institutions, the number of medical staff, the assets of medical resources, the per capita hospitalization expenses, and the insurance covering population in China have all shown a trend of increasing year by year [17], the utilization of medical services by migrant population is still at a low level [7]. After the outbreak of COVID-19, the sense of loneliness of elderly patients, especially females living alone, has intensified, their mental health has deteriorated significantly, and the utilization rate of medical services has decreased accordingly [18]. To increase the utilization of medical services for the rural population, it is suggested that rural residents’ ability to use mobile payment [19] and the social security system [20] shall be improved. On the contrary, excessive utilization of medical services is accustomed in many countries, posing a potential threat to public budgets and public health, especially in lower-income countries [21].

Regarding the relationship between public health education and medical service utilization, Gui [22] pointed out that both show a low level in Northeast China. It is agreed that receiving public health education is conducive to improving people’s awareness of autonomy in nursing so as to achieve medical care according to their needs [23]. Morevoer, when Sang et al. [24] examined the factors affecting the willingness of individuals to use primary healthcare services, they proposed that strengthening health education on common epidemics could improve the utilization efficiency of primary medical services. Furthermore, improving health education and publicity is an important factor that affects the medical utilization level of the migrant population [25]. Lack of health knowledge and parents’ active participation in children’s development significantly impact children’s health care to a certain extent [26].

The existing literature provides a good foundation for further study but shows little attention to the relationship between public health education and migrant workers’ medical service utilization. This paper uses 2017 CMDS data to explore such relations. The main marginal contributions are as follows. First, based on the unique perspective of public health education, this paper discovers the influencing factors of migrant workers’ medical service utilization, which may enrich the research on public health services. Second, this study takes the number of provincial community service centers and the average acceptance rate of public health education as the instrumental variables of public health education to conduct an empirical analysis of migrant workers’ medical service utilization behavior, making the estimation results more reliable. Third, this paper not only analyzes the heterogeneity of public health education on migrant workers’ medical service utilization but also tries to put forward the impact logic of migrant workers’ health literacy, social network, and psychological integration to verify the transmission mechanism of public health education on medical service utilization.

### 1.3. Analytical Framework

The World Health Organization defines “health literacy” as the ability of individuals to obtain, understand and practice health information and services, and use this information and services to improve their health [27]. As research indicates, health education plays a key role in improving health literacy [28] to improve population health. Specifically, a lack of health knowledge may lead to insufficient use of disease-prevention resources such as vaccination and routine screening [29]. Higher health literacy will positively promote the use of basic public health services for the migrant population [30]; that is, there is a correlation between individual health literacy and medical service utilization behavior. Therefore, this study assumes that public health education has an indirect impact on the utilization of medical services by improving the individual health literacy of migrant workers.

Besides, basic public health education has a certain impact on the social network and social integration of the migrant population [31]. With the acceptance of the concept of health and equity, the integration barriers of the migrant population will be eliminated to a certain extent [32], which is conducive to building an individual’s social network system. Moreover, mutual-aid behavior in social networks is related to individual health concepts, awareness, and health status [33]. Furthermore, through public health education, the migrant population will learn more about the local medical and health policies and maintain good health conditions. This allows them to participate in local activities and ultimately form a stable social network relationship. As studies found, with the promotion of social integration, migrant workers’ medical service utilization level is becoming better [34]. That is, when the migrant population gets more familiar with the local residents and the local medical and health environment, they shall take more change to use these resources.

In addition to building a good social network system, receiving public health education can significantly improve the psychological integration of the younger generation of the migrant population in urban adaptation, identity, and other aspects [35]. The higher the psychological integration of the migrant population, the better the integration level with urban communions, the sharing of medical service resources, and the removal of the differences in social cognition with local residents [36]. The practical social integration includes a high psychological identity of the migrant population with their settlement city [37], thereby intensifying the use of medical and health resources and improving their health level [38], which shows the impact on their medical service utilization. On the one hand, the community’s public health education, through collective education, provides an opportunity for the migrant and the local to communicate on an equal footing, changes their psychological and behavioral bias to a certain extent, improves migrant workers’ psychological adjustment ability, and enables them to better integrate into urban life. On the other hand, the higher the degree of psychological integration, the higher the migrant workers’ recognition of the living environment and medical services, and the greater the possibility of the decision to go clinic and hospital in need.

To sum up, this paper proposes that public health education can have an indirect impact on the utilization of migrant workers’ medical services through the intermediary role of improving health literacy, building social networks, and promoting psychological integration. The analytical framework is shown in Figure 1.

## 2. Methodology

### 2.1. Samples

The data in this paper comes from the survey data of dynamic monitoring of the migrant population in 2017, released by the National Health Commission of China, referred to as CMDS2017. It is an authoritative and representative survey to understand the survival and development status of the migrant population, the trend and characteristics of the migrancy, the utilization of public health services and family planning services of the migrant, etc. The data were collected from 31 provinces (incl. autonomous regions and municipalities) of China, from the migrant population aged 15 years and above (who stayed in the in-flow area for month at least). The total number of samples was 169,989. As this paper studies the medical service utilization of migrant workers and other issues, from this perspective, there is no age limit; therefore, to ensure the comprehensiveness of the samples, the research sample is defined as a “migrant population with agricultural household registration within the age of 15 to 70”. After screening, a total of 132,555 valid samples were finally obtained. 

### 2.2. Variables

The explained variable of this paper is the medical service utilization of migrant workers. Based on CMDS2017 data, choosing “where did you go to see the disease/injury first when you were sick (injured) or ill” as the quantitative indicator of medical service utilization, the study assigns a value of “1” to migrant workers who go to local personal clinics, community hospitals and general hospitals for medical treatment, and a value of “0” to whom do not go to any of the above institutions, and generates a discrete binary variable of medical service utilization of migrant workers.

This paper also compares the medical service utilization rate of migrant workers with different education levels, income levels, and occupations. Firstly, although there are differences in education levels, the overall probability of migrant workers choosing to see a doctor when ill is close to 50%. Specifically, 52.98% for primary school and below, 51.21% for junior high school, 50.92% for senior high school/technical secondary school, and 49.96% for college and above. Secondly, from the view of different income levels, we found that whether the average monthly income of migrant workers is less than 5000 yuan, 5000 yuan–10,000 yuan, or more than 10,000 yuan, the probability of using medical services is close to 52%. Finally, we divide migrant workers’ occupations into four categories: public officials, business and service personnel, agricultural practitioners, and professional technicians. Through comparison, it is found that agricultural practitioners have the highest probability of using medical services when ill, which is 53.05%, while the probability of public officials is relatively low. To sum up, migrant workers’ use of medical services does not show significant differences in education and income but in occupations.

In addition, this study divides the migrant population into the “new generation” (<40 years old) and the “old generation” (≥40 years old). Moreover, those who have moved for more than five years are classified as “long-term migrants”, and those less than or equal to five years are grouped as “short-term migrants”. The migrant workers who flow across provinces are classified as “trans-provincial migration”, and the others are the “non-trans-provincial migration”. Statistics show that no matter whether the new generation or the old generation, long-term immigrants or short-term immigrants, trans-provincial migration or non-trans-provincial migration groups, after receiving public health education, the probability of using medical service utilization is expanding, which to some extent indicates that migrant workers receiving public health education can promote their medical service utilization behavior. After receiving health education, the proportion of the new generation and the old generation of migrant workers who decide to see a doctor is equal, but the long-term migrants are nearly 2% higher than the short-term migrants in the use of medical services, the trans-provincial group is also nearly 2% higher than the non-trans-provincial, which indicates that the longer the migrant workers’ migration year, the better the public health education will be for the use of medical services, so is the migration distance.

The explanatory variable of this paper is public health education. According to the CMDS2017 questionnaire, choosing the question “in the past year, have you received health education on occupational disease prevention and control, STD and AIDS prevention and control, reproductive health and contraception, tuberculosis prevention and control, smoking control, mental health, chronic disease prevention and control, maternal and child care, and self-help in public emergencies in your current village (community)” as the indicator, and assign with a minimum value of 0 and a maximum value of 9 to respondents who have received the above health education. The average value of migrant workers receiving public health education is 3.7055; they received 0–9 kinds of public health education, accounting for 27.34%, 9.86%, 9.94%, 7.82%, 6.48%, 5.68%, 5.09%, 5.45%, 6.68%, and 15.65%, respectively. Among them, 27.34% of migrant workers have not received any public health education, indicating that the current coverage of public health education is still low.

The control variables in this study are selected from personal, family, employment, and migrancy characteristics. Personal characteristics include age, gender, education level, marital status, and health status. Family characteristics include the number of family members, family monthly income, whether the family has contracted land, and other variables. Employment characteristics include self-employed or not, working time per week, medical insurance covered or not, etc. The migrancy characteristics include the number of migrancy, migrating across provinces or not, and other variables. Considering the differences in policy effects in various regions, this study controls the regional effects as dummy variables. See Table 1 for descriptive statistics of specific variables.

### 2.3. Regression Method

The variable of medical service utilization of migrant workers in this study belongs to a binary classification variable. This study uses the Probit Model to test the impact of public health education on it. The regression model is as follows.
(1)$$MSU^{*}=\alpha +\beta_{i}HE_{i}+\gamma_{i}X_{i}+\varepsilon_{i}$$

In this equation, $MSU^{*}$ represents the latent variable of migrant workers’ medical service utilization, and $HE_{i}$ is the core explanatory variable, showing the public health education that migrant workers receive. $X_{i}$ is the control variable, including personal characteristics, family characteristics, socioeconomic characteristics, migrancy characteristics, etc. α is a constant. $\beta_{i}$ and $\gamma_{i}$ is the parameter to be estimated. $\varepsilon_{i}$ is the random interference. Moreover, in order to make the results more robust, this paper uses the variable segmentation method to verify the robustness of the impact of different types of public health education on the use of medical services.

As there may be endogenous problems such as missing variables, the direct use of the Probit Model may lead to errors in the regression results. This study uses the instrumental variable method based on IVProbit Model to solve these endogenous problems. The instrumental variable adopts the interactive term of the average number of “provincial community service centers” and the number of migrant workers receiving public health education in the fourth quarter of 2017. From a macro perspective, the more community service centers in a region, the better the effect of public health education. Research shows that the improvement of the basal health education service system can promote the effective implementation of the Health China Strategy [2]. To some extent, this shows that the number of community service centers at the basal level is closely related to public health education. In addition, the individual medical use behavior of migrant workers has little impact on the number of social service centers in each province, and the exogenous nature of this variable is established. From a micro perspective, migrant workers are subdivided into 1980 groups according to their communities to calculate the average value of the sub-groups receiving public health education. As a variable supporting a higher regional level variable, this variable is highly related to the explanatory variables in this paper. However, the utilization of individual medical services of migrant workers has less impact on the average value that they receive public health education at the group level. Therefore, this variable is exogenous. 

The IVProbit Model is as follows.
(2)$$\;HE_{i}=I*\left(\delta_{i}Z_{i}+\theta_{i}X_{i}+\tau_{i}\right)$$
(3)$$\;MSU^{*}=I*\left(\omega_{i}Z_{i}+\varphi_{i}HE_{i}+\varepsilon_{i}X_{i}+\sigma_{i}\right)$$

In Equations (2) and (3), I∗ represents characteristics function. When $\delta_{i}Z_{i}+\theta_{i}X_{i}+\tau_{i}$ > 0, it is assigned with value 1, otherwise it is assigned 0. $Z_{i}$ represents instrumental variables. $\tau_{i}$ and $\sigma_{i}$ are both random interferences and meet $Cov(\tau_{i},\sigma_{i})\neq 0$.

In order to increase the robustness of the regression results, this study replaces the dependent variables with cold treatment and health evaluation. The specific questions in the questionnaire are “do you have a cold treatment” and “have you ever received follow-up assessment, physical examination and other services for certain diseases provided by local community health service centers or township health centers free of charge”. The study assigns 1 to the “Yes” answer, and 0 to “No”, which are defined as new medical service utilization variables. In addition, this study replaced explanatory variables. It defines “whether the community has established a health file for the individuals” as a new explanatory variable, with value 1 assigned to “Yes”, and 0 assigned to “No”. Then, IVProbit Model is used to conduct regression tests on the replacement variables.

When dealing with the problem of respondents’ selection error and sample selection error, many studies use the propensity score matching method to simulate group samples to obtain the average treatment effect. However, traditional matching methods only focus on the propensity score, which cannot guarantee the combination and balance of all covariates. Therefore, Hainmueller [39] proposed the Entropy Balancing method. This method creates balanced samples in the data analysis of binary processing, which mainly imposes a set of moment constraints on the covariates of the control group to ensure the covariates of the processing group and the control group achieve accurate data matching under a specific moment, and automatically calculates a set of optimal weights matching the constraints. Then, the re-weighted data are used for regression analysis in the subsequent steps. As this method is applicable to discrete binary variables, this paper reprocesses the independent variables, assigns “1” to any three and more, and “0” to less than three of the health education items in “those who have received occupational disease prevention and control, STD and AIDS prevention and control, reproductive health and contraception, tuberculosis prevention and control, smoking control, mental health, chronic disease prevention and control, maternal and child health care and eugenics, and self-rescue education in public emergencies, in the past 3 years”.

In order to explore whether other factors affect the impact of public health education on migrant workers’ medical services, the intermediary effect model proposed by Baron and Kenny [40] has been used. On this basis, this study adopts instrumental variables to conduct the IVProbit regression test to reduce the regression error of the intermediary effect model. The process could be divided into three steps: first, check whether public health education affects the utilization of medical services; second, judge the influence of public health education on intermediary variables; third, test the impact of public health education and intermediary variables on the utilization of medical services. The formulas are as follows.
(4)$$\;MED_{i}=\rho_{i}HE_{i}+\partial_{i}X_{i}+\vartheta_{i}$$
(5)$$MSE_{i}=\omega_{1}HE_{i}+\omega_{2}MED_{i}+\omega_{i}X_{i}+\pi_{i}$$

Here, $MED_{i}$ represents the intermediary variable in this study, $\vartheta_{i}$ and $\pi_{i}$ represents the random error term. If $\rho_{i}$ of Formula (4), and $\omega_{1}$ and $\omega_{2}$ of Formula (5) show significance, the selected intermediary variables are appropriate, and the intermediary effect is established.

## 3. Results

### 3.1. Baseline Regression

Table 2 represents the results of the baseline regression. The Probit Model, it shows that public health education significantly increases the utilization of migrant workers’ medical services at the 1% confidence level. Dealing with the endogenous problem, the results based on IVProbit Model show that public health education can still significantly improve migrant workers’ medical service utilization behavior, and the coefficient value even becomes larger, which indicates that the original model may underestimate the impact of health education without addressing endogeneity. Then, the regression results show that migrant workers’ acceptance of public health education can promote their medical service utilization to a certain extent.

From an economic perspective, every additional public health education received by migrant workers will bring increase by 0.0237 in medical service utilization. In the survey sample, the average level of migrant workers’ medical service utilization is 0.5136, and the standard deviation is 0.4998. By calculating the economic effect of the IVProbit Model, it is found that for each additional type of public health education received, the level of migrant workers’ medical service utilization will increase by about 2.3%. From the perspective of human capital, receiving public health education could enlarge the utilization of migrant workers’ medical services, andincrease their health capital, so to improve their human capital stock. Therefore, this 5.4% growth in medical service utilization is of great significance for improving Chinese human capital.

In terms of control variables, based on the IVProbit Model, age is significant, and the value is small in personal characteristics. The health status variable is significantly positive, indicating that with the relative deterioration of health, migrant workers’ medical service utilization behavior becomes higher, which is consistent with the actual situation. Regarding to employment characteristics, working time shows that the longer the migrant workers work a week, the lower their medical service utilization level, and the participation of their medical insurance is positive in developing the utilization of migrant workers’ medical services. In terms of medical (distance) accessibility, the further migrant workers are away from the medical site, the less likely they are to choose medical services. Regarding migrancy characteristics, the coefficient value of interprovincial migrancy is significantly positive, indicating that interprovincial migrant workers have higher use of medical services, which means interprovincial migrancy impacts their physical and mental health to some extent. A previous study has confirmed that migrancy pressure, social capital, and other factors impact the mental health of interprovincial migrant workers [41].

### 3.2. Robust Test

In order to strengthen the robustness of the regression results, this study replaced currently explained variables with questions “do you want to see a doctor if you get a cold?” and “have you ever received follow-up assessment, physical examination and other services for certain diseases provided by local community health service center (station)/township health center free of charge?” in the questionnaire, with assigning the answer “yes” to 1 and “no” to 0, so to define them as the new medical service utilization variables. Table 3 shows the results of the robustness test. The regression results of replacing the explained variables reveal that migrant workers’ health education has a significant positive impact on cold treatment and physical evaluation, indicating that the result of public health education helping to promote the utilization of medical services for migrant workers is robust.

In this study, the current explanatory variable is replaced by the question “whether the community has established a health file for them” in the questionnaire, assigning the answer “yes” to value 1 and “no” to 0, so to generate a new explanatory variable. Table 3 shows that the coefficient of the alternative explanatory variable is significantly positive under the 5% confidence interval and indicates that the establishment of migrant workers’ public health archives has a significant positive impact on the utilization of medical services. As the migrant workers’ acceptance of public health education can affect their health status by establishing health records [15], the result verifies the robustness of the positive effect of public health education on the utilization of migrant workers’ medical services.

The control variables in this article are selected from the aspects of personal, family, employment and migrancy characteristics, which are used as covariates in the entropy balance matching method. After being weighted by the entropy balance method, the results showed that the mean, variance, and skewness of the covariates of migrant workers who received three or more public health education items and those who received less than three items were close. After matching, the regression results show that the public health education coefficient is still significantly positive, indicating it is robust that migrant workers’ acceptance of public health education can positively affect medical service utilization.

### 3.3. Heterogeneity Analysis

Due to the differences in people’s living environments and cultures, there are also great regional development differences. In this part, different migrant workers are grouped by age, education level, and city type to explore the heterogeneity of the impact of public health education on medical service utilization. The results are shown in Table 4. According to the standard of whether the respondents were born before 1980, the migrant workers are divided into the new generation and the old generation. Table 4 (A (Grouped with Age)) shows that the public health education of the old generation has significantly improved the utilization of medical services by the new generation. Table 4 (B (Grouped with Education)) is grouped according to different education levels. It shows that the groups with primary education have the least promotion effect, while the groups with secondary and higher education are significantly higher than those with primary education, and the difference between them is little. That is, the higher the education level is, the higher the promotion effect of public health education on migrant workers’ medical service utilization becomes. Table 4 (C (Grouped with City)) divides different city types according to the latest ranking of Chinese cities. The results show that the promotion effect of first-tier cities is significantly higher than that of second-tier cities.

The explanatory variable of this study is derived from the corresponding questions in the 2017 CMDS questionnaire, “have you received health education on occupational disease prevention and control, STD/AIDS prevention and control, reproductive health and contraception, tuberculosis prevention and control, smoking control, mental health, chronic disease prevention and control, maternal and child health care/prenatal care, and self-rescue in public emergencies in your current village/residence?”. In order to understand the difference in the impact of health education on migrant workers’ medical service utilization, this study subdivides the explanatory variables into five types of health education, occupational disease prevention, AIDS prevention, tuberculosis prevention, and chronic disease prevention are unified as “disease education”, and reproductive health and contraception, maternal and child health care and eugenics are unified as “reproductive education”. The regression results are shown in Table 5. It can be seen that different types of public health education affect migrant workers’ medical service utilization behavior. Among them, smoking control education has the greatest impact on the utilization of migrant workers’ medical services, while disease education has the lowest. The effects of other education types from high to low are reproductive education, emergency education, and mental health education.

Furthermore, this study conducts a classified analysis of disease education. The regression results in Table 6 show that migrant workers’ occupational disease education has the greatest impact on their medical service utilization, while STD/AIDS prevention education has the lowest. Their migrancy can be attributed to survival pressure and employment choice. Therefore, compared with other types of disease education, occupational disease education in communities can attract more attention from migrant workers. As for the popularization of STD and AIDS education, migrant workers do not have sufficient awareness of it, and because of the shackles of traditional ideas, they try to escape the discussion of this topic, resulting in low utilization of their medical services.

Combined with Table 5 and Table 6, it shows that the community smoking control education has made most migrant workers aware of the dangers of smoking and built up a sense of prevention in terms of mental health, occupational disease prevention, and self-rescue in emergencies, which significantly promote the use of medical services. However, the publicity of the prevention of STDs, AIDS, and tuberculosis is not enough, and migrant workers’ awareness of the utilization of medical resources needs to be improved. Nevertheless, there is still much room for development in public health education. The communities need to improve the publicity method in all aspects.

## 4. Discussion and Conclusions

### 4.1. Discussion

The above regression results have already shown that migrant workers’ medical service utilization could be improved with the publicity of public health education, while the path of how public health education affects migrant workers’ utilization of medical services has not been discussed. In this regard, this study will analyze the impact path in three ways, health literacy, social network, and psychological integration.

Based on the question “have you heard of the national public health service project” in the questionnaire, this study set a proxy variable of migrant workers’ health literacy accordingly. The total of those who heard of the national project is about 58.52%, indicating that the current health awareness and health literacy of migrant workers need to be strengthened. For social networks, this study selects two proxy variables from the questionnaire, namely, “have you ever participated in trade unions/volunteer associations/homecoming associations/fellow villagers’ associations/hometown chambers of commerce/other activities?” and “will you report relevant situations to the government departments in various ways/put forward policy recommendations” to approximately reflect the social network relationship of migrant workers. For psychological integration, this study selected two questions, “do you agree with the statement that I am already a local” and “do you agree with the statement that local people look down on outsiders” as indicators. The article assigns 1 to the answer “fully agree” and “basically agree”, and 0 to “disagree” and “completely disagree”, Table 7 is hereby formulated. The IVProbit Model shows that migrant workers receiving public health education can significantly improve their health literacy, expand their social network and deepen their psychological integration level.

Table 8 shows the impact of public health education and intermediary variables on migrant workers’ medical service utilization. As a measure of health literacy, understanding the national health project can improve the medical service utilization of migrant workers. Meanwhile, the intermediary variable of the social network significantly affects their medical service utilization behavior. Moreover, the higher the level of psychological integration, the higher the medical service utilization of migrant workers. Public health education can promote the migrant workers’ use of medical services by improving their health literacy, social network expansion, and psychological integration. This conclusion is consistent with the elaboration of the impact path proposed in this article.

### 4.2. Conclusions

This article analyzes the impact of public health education on migrant workers’ medical service utilization by using the dynamic monitoring survey data of the migrant population in 2017. The main results show that after controlling personal, family, employment, and migrancy characteristics, receiving public health education can significantly improve their medical service utilization behavior. After using the instrumental variable method, expanding dependent variable and independent variable indicators, and using the entropy balance matching method to reduce bias, the research results are still robust. The heterogeneity analysis found that the public health education of the older generation and those in the first-tier cities had a higher effect, and the groups receiving secondary education and higher education were higher than those receiving primary education. The medical service utilization behavior of migrant workers receiving different types of public health education is also different, among which smoking control education produces the most obvious effect on medical service utilization. Regarding education on various disease types, receiving occupational disease education has the highest promotion effect on migrant workers’ medical service utilization, while receiving STD/AIDS education has the lowest. The intermediary variable test shows that health literacy, social network, and psychological integration are important ways for public health education to increase medical service utilization.

Therefore, the implications of this study are as follows. First, to improve the utilization level of migrant workers’ medical services, public health education is a considerable way. Governments should try to organize health education service institutions to cultivate migrant workers’ health concepts and literacy, change their incorrect health awareness and behavior, and set up health archives for them. Second, to promote the equalization of public health services, to improve the availability of public health education, and to broaden the psychological integration level of migrant workers through the cultivation of social networks and social capital. Governments shall take health education as an important part of basic education, and develop individuals’ health literacy, to improve the use of public health services. Third, as promoting public health education has a certain relationship with community governance, the community should strengthen the popularization of knowledge about AIDS, STDs, and other infectious diseases and pay special attention to the public health education activities of the younger generation, those with lower education level and those from non-first-tier cities.

Based on the perspective of public health education, this paper explores the utilization of migrant workers’ medical services and, to a certain extent, appeals to the state and government to pay attention to the health rights and interests of migrant workers. When exploring the heterogeneity of public health education in the use of medical services, we found that despite of different ages, different education levels, or living in different types of cities, receiving public health education can significantly promote the utilization of medical services by migrant workers, which proves the applicability of this study from the side. From the view of economics, China’s migrant workers have made great contributions to industrialization and urbanization. In the context of Healthy China, promoting timely medical treatment for them in case of illness and encouraging the initiative of using medical services are the inherent requirements of the Healthy China Strategy. However, due to the paradox of “economic acceptance—social exclusion”, the migrant population in a strange environment could not make full use of medical services with insufficient psychological integration, low health literacy, and an underdeveloped social network, to decide to go to the hospital when ill, and have regular physical examinations, etc. Therefore, this paper starts with the public health education of migrant workers, hoping that immigrants can receive more public health education and improves their psychological integration, health literacy, and social network to increase the medical service utilization efficiency. Nevertheless, this article uses cross-sectional survey data, which has certain limitations in the vertical dimension; in this regard, the authors will continue to pay attention to the use of dynamic tracking data for follow-up research.

## Figures and Tables

**Figure 1 ijerph-19-15879-f001:**
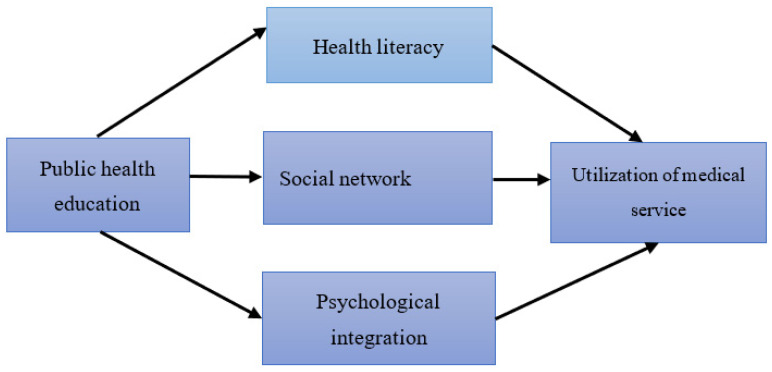
Analytical framework.

**Table 1 ijerph-19-15879-t001:** Descriptive statistics of variables.

Variables	Variable Definition	Mean	SD
**Explained Variable**			
Utilization of medical service	For whether go to clinic, community hospital and general hospital when ill. Yes = 1, No = 0	0.5136	0.4998
**Explanatory Variable**			
Public health education	Respondents receiving public health education. Min = 0, Max. = 9.	3.7055	3.3740
**Intermediary variable**			
Health literacy	For whether know about national health projects. Yes = 1, No = 0	0.5852	0.4927
Social network			
- Political participation	For whether report/advise to governments. Yes = 1, No = 0	0.3929	0.4884
- Activity participation	For whether join in local activities. Yes = 1, No = 0	0.4240	0.4942
Psychological integration			
- Admit to be native	Yes = 1, No = 0	0.7436	0.4366
- Recognized by local	Yes = 1, No = 0	0.8178	0.3860
**Control variables**			
Age	Continuous variable	36.2940	10.7684
Gender	Male = 1, Female = 0	0.5199	0.4996
Marital status	Married = 1, Unmarried = 0	0.9728	0.1626
Education level	Uneducated = 1, Primary = 2, Middle school = 3, High school = 3, Vocational school = 4, University = 5, Postgraduate = 6	3.2297	1.0363
Health status	Healthy = 1, Basically healthy = 2, Unhealthy = 3, Disable = 4	1.2069	0.4722
Number of family member	Continuous variable	3.2004	1.2133
Contracted land	Have = 1, Not have = 0	0.5777	0.4939
Monthly income	Continuous variable	8.6354	0.5918
Employment type	Self—employed = 1, Not self—employed = 0	0.4247	0.4943
Working time	Continuous variable (per week)	0.9408	0.2361
Medical insurance	Covered = 1, Not covered = 0	0.1597	0.3663
Medical access	1–4 Ordered variable	1.1892	0.4534
Monthly income	Continuous variable	2.0184	1.9614
Employment type	Interprovincial = 1, Not = 0	0.5106	0.4999

**Table 2 ijerph-19-15879-t002:** The impact of public health education on migrant workers’ medical service utilization.

Variables	Probit Model	IVProbit Model
Public health education	0.0204 ***	0.0237 ***
	(0.00203)	(0.00751)
Age	−0.00418 ***	−0.00418 ***
	(0.000784)	(0.000788)
Gender	−0.00918	−0.00809
	(0.0132)	(0.0134)
Marital status	0.0868 **	0.0918 **
	(0.0415)	(0.0423)
Education level	−0.0249 ***	−0.0265 ***
	(0.00771)	(0.00797)
Health status	0.171 ***	0.173 ***
	(0.0136)	(0.0140)
Number of family member	0.0246 ***	0.0240 ***
	(0.00692)	(0.00696)
Contracted land	−0.0155	−0.0148
	(0.0138)	(0.0139)
Monthly income (logarithm)	0.00528	0.00439
	(0.0131)	(0.0131)
Employment type	0.00776	0.00751
	(0.0148)	(0.0149)
Working time	−0.00215 ***	−0.00212 ***
	(0.000375)	(0.000377)
Medical insurance	0.0517 ***	0.0534 ***
	(0.0184)	(0.0188)
Medical access	−0.0411 ***	−0.0368 **
	(0.0155)	(0.0157)
Migrancy times	−0.00183	−0.00213
	(0.00329)	(0.00326)
Interprovincial	0.0429 ***	0.0428 ***
	(0.0155)	(0.0155)
Provincial effect	Y	Y
Constant	−0.275 **	−0.283 **
	(0.136)	(0.138)
Observations	39,841	39,310

Robust standard error in brackets. *** *p* < 0.01, ** *p* < 0.05.

**Table 3 ijerph-19-15879-t003:** Robust test.

Variables	Replacement of Explained Variable		Replacement of Explanatory Variable	Entropy Balance Matching Method
	Cold Clinic	Physical Assessment	Medical Service Utilization	Medical Service Utilization
Public health education	0.0221 ***	0.148 ***		0.0896 ***
	(0.00638)	(0.0225)		(0.0129)
Public health file			0.332 ***	0.0349
			(0.0925)	(0.0640)
Control variable				
Constant	−0.428 ***	−1.507 ***	−0.287 *	−0.225 *
	(0.130)	(0.452)	(0.150)	(0.132)
Observations	46,503	4023	33,705	42,976

Robust standard error in brackets. *** *p* < 0.01, * *p* < 0.1. Control variables are the same with Table 2.

**Table 4 ijerph-19-15879-t004:** Heterogeneity of public health education on medical service utilization of different migrant workers.

	A (Grouped with Age)		B (Grouped with Education)			C (Grouped with City)	
Variables	Old Generation	New Generation	Primary Education	Middle Education	Higher Education	First-Tier	Second-Tier
Public health education	0.0192 ***	0.0217 ***	0.0153 ***	0.0215 ***	0.0206 ***	0.0239 ***	0.0105 **
	(0.00274)	(0.00305)	(0.00505)	(0.00243)	(0.00616)	(0.00351)	(0.00458)
Control variable	Ctrl	Ctrl	Ctrl	Ctrl	Ctrl	Ctrl	Ctrl
Constant	−0.115	−0.520 **	−0.886 ***	−0.0989	0.156	−0.609 ***	0.434
	(0.188)	(0.213)	(0.323)	(0.173)	(0.469)	(0.231)	(0.315)
Observations	21,584	18,257	6911	27,761	4131	14,214	8427

Robust standard error in brackets. *** *p* < 0.01, ** *p* < 0.05. Control variables are the same as in Table 2.

**Table 5 ijerph-19-15879-t005:** Classification based on different types of public health education.

	(1)	(2)	(3)	(4)	(5)
Disease education	0.180 ***				
	(0.0617)				
reproductive education		0.223 ***			
		(0.0763)			
Smoking control			0.202 ***		
			(0.0640)		
Mental education				0.213 ***	
				(0.0676)	
Emergency education					0.221 ***
					(0.0700)
Control variables	Ctrl	Ctrl	Ctrl	Ctrl	Ctrl
Constant	−0.294 **	−0.323 **	−0.314 **	−0.287 **	−0.273 **
	(0.131)	(0.133)	(0.140)	(0.139)	(0.138)
Observations	42,400	42,400	39,310	39,310	39,310

Robust standard error in brackets. *** *p* < 0.01, ** *p* < 0.05. Control variables the same as in Table 2.

**Table 6 ijerph-19-15879-t006:** Classification based on different types of disease education.

	(1)	(2)	(3)	(4)
Occupational disease education	0.229 ***			
	(0.0726)			
STD/AIDS prevention education		0.191 ***		
		(0.0629)		
Tuberculosis education			0.209 ***	
			(0.0661)	
Chronic disease education				0.195 ***
				(0.0616)
Control variables	Ctrl	Ctrl	Ctrl	Ctrl
Constants	−0.257 *	−0.272 **	−0.255 *	−0.264 *
	(0.137)	(0.138)	(0.137)	(0.137)
Observations	39,310	39,310	39,310	39,310

Robust standard error in brackets. *** *p* < 0.01, ** *p* < 0.05, * *p* < 0.1. Control variables same with Table 2.

**Table 7 ijerph-19-15879-t007:** The impact of public health education on intermediary variables.

	Health Literacy	Social Network		Psychological Integration	
Variables	National Health Project	Political Participation	Social Participation	National Health Project	Political Participation
Public health education	0.227 ***	0.0789 ***	0.0875 ***	0.0573 ***	0.0322 ***
	(0.00519)	(0.00483)	(0.00481)	(0.00504)	(0.00533)
Control variables	Ctrl.	Ctrl.	Ctrl.	Ctrl.	Ctrl
Constants	−1.129 ***	−2.553 ***	−2.600 ***	−0.255 **	0.368 ***
	(0.103)	(0.0989)	(0.0996)	(0.107)	(0.111)
Obsercations	79,654	79,654	79,654	79,654	79,654

Robust standard error in brackets. *** *p* < 0.01, ** *p* < 0.05. Control variables same with Table 2.

**Table 8 ijerph-19-15879-t008:** The impact of public health education and intermediary variables.

	(1)	(2)		(3)	
Variables	Health Literacy	Social Network		Psychological Integration	
National health project	0.122 ***				
	(0.0129)				
Political participation		0.102 ***			
		(0.0128)			
Social participation			0.114 ***		
			(0.0129)		
Admit oneself to be local				0.0773 ***	
				(0.0141)	
Be accepted by the local					0.00208
					(0.0155)
Control variables	Ctrl.	Ctrl.	Ctrl.	Ctrl.	Control variables
Constants	−0.276 **	−0.212	−0.207	−0.268 **	−0.241 *
	(0.129)	(0.130)	(0.129)	(0.129)	(0.130)
Observations	42,976	42,976	42,976	42,976	42,976

Robust standard error in brackets. *** *p* < 0.01, ** *p* < 0.05, * *p* < 0.1. Control variables same with Table 2.

## Data Availability

The data that support the findings of this study are available from the corresponding author, H.W. upon reasonable request.

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
