# Peer review of "The Impact of Public Health Education on Migrant Workers’ Medical Service Utilization"

_ijerph, 2022, doi:10.3390/ijerph192315879_

Round 1

Reviewer 1 Report

The proposed paper ‘The impact of public health education on migrant workers' medical service utilization’ analyzes the relationship between public health education and the use of medical services in the population of working migrants in China. The analysis is based on a regression model (adjusted for other factors), whose target and explicative variables are both derived from a self-compiled questionnaire.

The methodologic framework is scientifically sound and formally correct, however there are some issues that, in my opinion, require attention.

1) The only knowledge improvement offered by the study is in the assessment of the target hypothesis, hence the fact that public health education does increase the use of health services in a specific target population. However, it is unclear why this fact should be considered of interest.

Authors state: ‘From the perspective of human capital, receiving public health education could enlarge the utilization of migrant workers' medical services, increase their health capital, so to improve their human capital stock. Therefore, this 5.4% growth in medical service utilization is of great significance for improving Chinese human capital.’ From this statement, it appears that authors consider the use of health services as a good to be pursued. However, the use of health services is not a good in itself: it is so when it generates an improvement of health conditions, an aspect that is not addressed in this study. How can authors be sure that the reported use of health services had a positive (and otherwise unachievable) impact on population health? On the opposite, an unmotivated use of health services is undesirable, as it should be considered a waste of resources.

Moreover, there is no discussion about the economic implications of health services use. Is the access to the all the object healthcare services free to everybody? If it is not the case, a higher use of health services (possibly unmotivated by health needs, as this aspect is not assessed) can be seen as an additional taxation, possibly on populations living in lower conditions. Another possible case is that the use of such services is only guaranteed under employment-related assurance, but also in this case a deeper analysis on the status of the target population should be included.

Authors state that increasing the level of use of health services by the target population is within the scopes of the ‘Health China Strategy (Sun, 2021)’, but this cannot be considered as a universal aim to be achieved, once again (see comment 1) because the impact on population health is not addressed. Therefore, the scope of investigating the target hypothesis, and the relevance of the results, is unclear.

2) It is unclear what is the possible appeal of the proposed study to the international scientific community. The characteristics of the target population are not specified, and therefore it is impossible to assess if the results of the study can be considered valid for other territories outside of China. It is necessary to present a deeper description and analysis of the target population, and of the social context in which they live, in order to assess if the identified cause-effect mechanism is likely to be valid in different geographical and social contexts.

3) An official ethical approval is missing to use human-generated personal data for the proposed study.

Other minor comments are:

4) The first three chapters (1. Background, 2. Literature Review, 3. Analytical Framework) should be sections of a single chapter 1. Introduction. Also 6. Discussion and 7. Conclusions should be unified.

5) Line 219-220 ‘for theoretically the more community service centers in a region, the better the effect of public health education.’: this statement is unclear. Authors state that a higher number of community service centers improves the effect of public health education; however, a higher number of community service centers should facilitate the access to health services, and therefore a higher level of use of such services is to be expected, regardless its relation with public health education. It is otherwise likely that a higher number of community service centers may facilitate knowledge spreading, and therefore can improve the level of public health education, but it is unclear why it should be evident (as authors state) that it also enhances the impact of public health education on health services use. Please elaborate. This is particularly relevant as this is presented as the reason why authors chose it as the instrumental variable for the IVProbit Model.

6) Line 273: the terms p1 and p2 are not present in the formula, where only pi  is present. Do authors mean w1 and w2?

7) Line 466 ‘establishment of public health archives and other aspects.’ ’ Possibly a typo.

Author Response

Review 1:

1) The only knowledge improvement offered by the study is in the assessment of the target hypothesis, hence the fact that public health education does increase the use of health services in a specific target population. However, it is unclear why this fact should be considered of interest.

Authors state: ‘From the perspective of human capital, receiving public health education could enlarge the utilization of migrant workers' medical services, increase their health capital, so to improve their human capital stock. Therefore, this 5.4% growth in medical service utilization is of great significance for improving Chinese human capital.’ From this statement, it appears that authors consider the use of health services as a good to be pursued. However, the use of health services is not a good in itself: it is so when it generates an improvement of health conditions, an aspect that is not addressed in this study. How can authors be sure that the reported use of health services had a positive (and otherwise unachievable) impact on population health? On the opposite, an unmotivated use of health services is undesirable, as it should be considered a waste of resources.

Moreover, there is no discussion about the economic implications of health services use. Is the access to the all the object healthcare services free to everybody? If it is not the case, a higher use of health services (possibly unmotivated by health needs, as this aspect is not assessed) can be seen as an additional taxation, possibly on populations living in lower conditions. Another possible case is that the use of such services is only guaranteed under employment-related assurance, but also in this case a deeper analysis on the status of the target population should be included.

Authors state that increasing the level of use of health services by the target population is within the scopes of the ‘Health China Strategy (Sun, 2021)’, but this cannot be considered as a universal aim to be achieved, once again (see comment 1) because the impact on population health is not addressed. Therefore, the scope of investigating the target hypothesis, and the relevance of the results, is unclear.

Authors: Thank the reviewer for the suggestions. As the reviewer put forward, this article needs to demonstrate the effectiveness of medical service utilization. We would explain it from the following two aspects. On the one hand, this paper distinguishes the difference in the impact of public health education on the utilization of medical services under different health conditions, and finds that the impact of public health education is significantly positive in the groups with poor health and those with good health condition. On the other hand, this paper divides the samples into low-income, middle-income and high-income groups to test the impact of public health education on the use of medical services on different economic groups. The results show that public health education has a significant positive impact especially on the low-income groups. Therefore, as the reviewers put forward, the utilization of medical services may not be free, but the utilization efficiency is an important way to improve people's health. This paper makes a classified investigation from the group differences of health status and economic status, it shows that public health education can at least promote the fairness of medical service utilization among the poor health and low-income people from the side, which can respond to the reviewers' questions about the efficiency of medical service utilization from the perspective of health and economy to a certain extent.

Limited to the length of the article, it is only explained in the revision reply here, not listed in the text.

Table 1

classified by healthy status

classified by income

Healthy

Unhealthy

low

middle

high

Public health education

0.0196***

0.0247***

0.0204***

0.0233***

0.0207***

(0.00230)

(0.00438)

(0.00331)

(0.00256)

(0.00427)

Constant

-0.0916

0.634**

1.000***

0.407

-0.144

(0.153)

(0.302)

(0.297)

(0.324)

(0.411)

Observations

30,804

9,037

14,816

25,550

8,988

2) It is unclear what is the possible appeal of the proposed study to the international scientific community. The characteristics of the target population are not specified, and therefore it is impossible to assess if the results of the study can be considered valid for other territories outside of China. It is necessary to present a deeper description and analysis of the target population, and of the social context in which they live, in order to assess if the identified cause-effect mechanism is likely to be valid in different geographical and social contexts.

Authors: We thank the reviewer for pointing out our shortcomings in defining the target population. We have responded and revised them item by item according to the comments. About the question whether this topic is attractive enough for the international academy, we believe that we take China's public health education as an example to explore the basic situation of the medical service utilization of migrant workers, which may have some reference value for domestic and foreign scholars who are interested in China's health research. As for the insufficient description of the basic characteristics of the target population, we very much agree with the suggestions. Therefore, during revising, the background description is supplemented into the part of introduction, and the description of migrant workers' education, income, occupation and other aspects is added in the variable definition part, by that readers can understand this research more clearly.

3) An official ethical approval is missing to use human-generated personal data for the proposed study.

Authors: This study was approved by the Ethics Committee of Anhui University of Finance and Economics, and in accordance with the provisions of the Helsinki Declaration, the specific content of the study does not involve topics such as human genes, and only elaborates the impact of public health education on the use of medical services from a microeconomic perspective.

4) The first three chapters (1. Background, 2. Literature Review, 3. Analytical Framework) should be sections of a single chapter 1. Introduction. Also 6. Discussion and 7. Conclusions should be unified.

Authors: Thank the reviewer for the suggestions about the article framework. According to the comments, we have consolidated the relevant frameworks, and integrated the discussions and conclusions.

5) Line 219-220 ‘for theoretically the more community service centers in a region, the better the effect of public health education.’: this statement is unclear. Authors state that a higher number of community service centers improves the effect of public health education; however, a higher number of community service centers should facilitate the access to health services, and therefore a higher level of use of such services is to be expected, regardless its relation with public health education. It is otherwise likely that a higher number of community service centers may facilitate knowledge spreading, and therefore can improve the level of public health education, but it is unclear why it should be evident (as authors state) that it also enhances the impact of public health education on health services use. Please elaborate. This is particularly relevant as this is presented as the reason why authors chose it as the instrumental variable for the IVProbit Model.

Authors: The reviewer states that a higher community service center would help spread knowledge and improve the level of public health education. This shows that there is a correlation between instrumental variables and explanatory variables. Therefore, on the premise of meeting the relevance and externality, this paper re-adopts the IVProbit model to overcome the endogeneity. The study finds that public health education still has a positive and significant impact on the utilization of medical services after using instrumental variables to solve the endogeneity. What needs to be explained is that as the instrumental variable is related to the explanatory variable, but not to the disturbance item, the impact of the number of communities on the utilization of medical services is generated through public health education. As the reviewer pointed out, our original statement about the number of communities has enhanced the impact of public health education on the utilization of medical services, this impact logic is not the role of instrumental variables, but of the positive regulation. Our original statement may have some ambiguity and has been revised accordingly.

6) Line 273: the terms p1 and p2 are not present in the formula, where only pi is present. Do authors mean w1 and w2?

Authors: We are grateful for the suggestions. As the reviewer pointed out, the marks are w1 and w2, which have been modified in the text.

7) Line 466 ‘establishment of public health archives and other aspects. Possibly a typo.

Authors: We are grateful for the suggestions The details we have been modified accordingly.

Reviewer 2 Report

Thank you for the opportunity to peer review this important work looking at migrant health utilisation and public health education. 

This retrospective analysis using routine population data provides new evidence of the important of public health education in migrant population to optimise their healthcare utilisation.

I believe the manuscript requires revision, and I note suggestions below:

  1. Analytical Framework and implications to regression modelling. It is good to see that you provide the evidence around the analytical framework that the authors have chosen for your regression analysis and the relationship between the dependent and independent variable is clearly explained. It would also be beneficial to present the confounders  variables including those that you control for and you may consider to do a DAG for this - or present this as an appendix if you have done this already. 
  2. Confounders /control variables. I believe it would be beneficial to ascertain what all confounders are and which you are able to control for and which you cannot control for and why and list this as a limitation. I see the SES, age, gender variables are controlled for but I think it is worth noting that migrant healthcare utilisation may be determined by how far away they are from a hospital/healthcare setting and similarly the amount of public health education they receive may also be determine by where they live. I think geographical location is an example of a variable that would act as a confounder in your work and would need addressing. 
  3. Population. Looking at lines 167-170 in Methodology it appears that the inclusion sample is for ‘migrant population with agricultural household registration within the age of 15-70’. If this is part of the inclusion criteria it would be worth making this clear and listing the inclusion/exclusion criteria - a flowchart would be helpful to display this. 
  4. Similarly, a table summarising key characteristics of the study population would be useful to the reader to put the findings into context.
  5. P10 lines 362-375, where authors go onto explaining that the explanatory variable is subdivided into five types should be part of the methodology instead of the results.
  6. In the discussion, I would welcome discussion about migrant healthcare utilisation and what appropriate use of health services look like and how that could be looked at. 
  7. Limitations, when using routine data especially data from survey/self-reports it is important to discuss the limitations at length and I believe that would strengthen this manuscript. 
  8. Finally, details of future research questions would complement the policy implications that authors included. 

I believe that with the revision above, the findings of the analysis will become clearer to the reader and will provide enough information for them to make a decision as to how generalisable the findings are.

Author Response

Review 2

I believe the manuscript requires revision, and I note suggestions below:

  1. Analytical Framework and implications to regression modelling. It is good to see that you provide the evidence around the analytical framework that the authors have chosen for your regression analysis and the relationship between the dependent and independent variable is clearly explained. It would also be beneficial to present the confounders  variables including those that you control for and you may consider to do a DAG for this - or present this as an appendix if you have done this already. 

Authors: Thanks to the reviewer for the suggestions on adding DAG and appendix to the article. What needs to be explained to the reviewer is that, the intermediary variables in Figure 1 are introduced to build a logical framework between explanatory variables and explained variables, which can better present the logic ideas of the study. As for the appendix, the control variables in our paper are various, that control other explanatory variables in the subsequent steps, but are not presented one by one, by which other explanatory variables have better effects in the benchmark regression model. In this regard, we would like to thank the reviewers for putting forward this opinion, which has helped to improve the logical relationship between the variables.

  1. Confounders /control variables. I believe it would be beneficial to ascertain what all confounders are and which you are able to control for and which you cannot control for and why and list this as a limitation. I see the SES, age, gender variables are controlled for but I think it is worth noting that migrant healthcare utilisation may be determined by how far away they are from a hospital/healthcare setting and similarly the amount of public health education they receive may also be determine by where they live. I think geographical location is an example of a variable that would act as a confounder in your work and would need addressing. 

Authors: Thank you very much for the suggestions. We highly agree with your suggestions on the selection of control variables, especially controlling the medical distance or relevant geographical location. Therefore, we’d like to reply as followings. Firstly, we increase the control variable of the distance of migrant workers who choose medical service sites when they are ill, which is an ordered variable, of which "from your residence to the nearest medical service institution (including community health service center, village clinic, hospital, etc.) within 15 minutes=1, 15 minutes (excluding)- 30 minutes (including)=2, 30 minutes (excluding) - 1 hour (including)=3, more than 1 hour=4". Secondly, regarding to the geographical location variables, it should be explained that this study has conducted geographical location control from the province of residence (described in the variable definition) and the mobility range, which can mitigate the impact of the geographical location variables proposed by the reviewers to a certain extent.

  1. Population. Looking at lines 167-170 in Methodology it appears that the inclusion sample is for ‘migrant population with agricultural household registration within the age of 15-70’. If this is part of the inclusion criteria it would be worth making this clear and listing the inclusion/exclusion criteria - a flowchart would be helpful to display this. 

Authors: For reviewer’s question of the inclusion sample of ‘migrant population with agricultural household registration within the age of 15-70’. Our answer is as followings. Firstly, in terms of age selection, from the perspective of public health education and medical service utilization, it is sure that there is no age limit for the migrant workers. This study limits the age to 15-70, mainly to ensure the comprehensiveness of the sample to a certain extent. Secondly, in terms of the employment characteristics, the number of migrant workers is large, but their income is low (according to the relevant data of the National Bureau of Statistics, the number of migrant workers in China in 2017 was as high as 170 million, but the average monthly income was less than 3500 yuan). This has brought great economic pressure on migrant workers, who often has to choose to extend their working years. Moreover, Chinese residents may have a stronger sense of saving and choose to make adequate preparations for the future. Therefore, this paper does not select the legal working age as age definition, but expands the age range to 15 to 70 years old to ensure the accuracy of the sample. Limited to article length, this part is not detailed in the original text.

4.Similarly, a table summarizing key characteristics of the study population would be useful to the reader to put the findings into context.

Authors: We are grateful of the reviewer’s suggestion of summarizing key characteristics of migrant workers. We highly agree on it and believe that the improvement will make the study more convincing. Our revise is as follows. Firstly, we have supplemented relevant data on migrant workers in the introduction, which is also one of the reasons for choosing migrant workers as research samples. Second, we have supplemented the situation of migrant workers receiving public health education in detail, and specifically discussed the medical service utilization of migrant workers with different education, income and occupation, further increasing the discussion on the characteristics of migrant workers, so to make the article more complete.

5.P10 lines 362-375, where authors go onto explaining that the explanatory variable is subdivided into five types should be part of the methodology instead of the results.

Authors: Thank the reviewer for pointing out this question. We’ve corrected in the paper, and added a description of subdividing variables in the methodology.

6.In the discussion, I would welcome discussion about migrant healthcare utilisation and what appropriate use of health services look like and how that could be looked at. 

Authors: The reviewer pointed out that the applicability of medical service utilization should be added in the discussion. In this regard, we further tested the difference in the impact of public health on the utilization of medical services among groups with different health status and income levels, in combination with the opinions of another reviewer. For the poor health and low-income groups, they are the weak side of medical service utilization. If public health education can improve their medical service utilization level, it indirectly indicates that public health education has certain applicability to different groups. The specific results are shown in the following table, which is not presented in the text, due to the limit length of the article.

Table 1

classified by healthy status

classified by income

Healthy

Unhealthy

low

middle

high

Public health education

0.0196***

0.0247***

0.0204***

0.0233***

0.0207***

(0.00230)

(0.00438)

(0.00331)

(0.00256)

(0.00427)

Constant

-0.0916

0.634**

1.000***

0.407

-0.144

(0.153)

(0.302)

(0.297)

(0.324)

(0.411)

Observations

30,804

9,037

14,816

25,550

8,988

7.Limitations, when using routine data especially data from survey/self-reports it is important to discuss the limitations at length and I believe that would strengthen this manuscript. 

Authors: Thank the reviewer for pointing out the problem of research limitations. As revealed in the article, the migrant frequency of China's floating population is high, which makes it difficult to obtain large-scale dynamic tracking data. In this regard, this study uses cross-sectional survey data, though it has certain limitations in the vertical dimension. Nevertheless, we added the statement of this limitation end of the article and will keep pay attention to the follow-up research using dynamic tracking data.

8.Finally, details of future research questions would complement the policy implications that authors included. 

Authors: Thank the reviewers for putting forward the details of the policy implications. We further modified and adjusted the policy recommendations to make them more consistent with the research conclusions and enhance the future reference significance of the policy recommendations.

Round 2

Reviewer 1 Report

Dear authors,

Thank you for the time and the efforts invested in the review. The manuscript resulted improved from the first version, but I still have some doubts related to comments 1 and 2.

Comment 1):

I appreciate the differentiated analysis according to groups characteristics (in terms of health status and income condition). However, the results somehow corroborate my perplexity. Authors state that: 'the impact of public health education is significantly positive in the groups with poor health and those with good health condition',  meaning that also in the 'good health' groups there is an increase in health services utilization. In the case this is related to prevention protocols, the outcome is desirable, but this should be specified in the text; otherwise, this result can be seen as an access to medical care from people who don't need it, which is an undesirable outcome. The pursuit of an improvement in the three 'intermediary variables' (health literacy, social networking, psychological integration) is evidently desirable, but their conversion into a higher use of health services is not necessarily a good outcome, depending on the health status. I would appreciate if authors could elaborate this aspect in the main text a little bit more, clarifying the social and ethical scope of the study object.

Comment 2):

The details provided about the characteristics of the study population are still insufficient, considering that the scope is to compare this population with other potential populations in different geographical areas. Some necessary details are age distribution (median and quartiles), ethnic information, and timing information about the migration moment, i.e. years from the migration or whether they are first or second generation migrants.

Author Response

Dear reviewer,

Thank you for your second-round review and giving us such constructive comments. Here are our answers according to your comments (in blue). The revise is done accordingly in the manuscript.

Comment 1):

I appreciate the differentiated analysis according to groups characteristics (in terms of health status and income condition). However, the results somehow corroborate my perplexity. Authors state that: 'the impact of public health education is significantly positive in the groups with poor health and those with good health condition', meaning that also in the 'good health' groups there is an increase in health services utilization. In the case this is related to prevention protocols, the outcome is desirable, but this should be specified in the text; otherwise, this result can be seen as an access to medical care from people who don't need it, which is an undesirable outcome. The pursuit of an improvement in the three 'intermediary variables' (health literacy, social networking, psychological integration) is evidently desirable, but their conversion into a higher use of health services is not necessarily a good outcome, depending on the health status. I would appreciate if authors could elaborate this aspect in the main text a little bit more, clarifying the social and ethical scope of the study object.

Answer: We are grateful for the reviewer’s constructive suggestions, which is quite important for improving the research accuracy of our paper. In this regard, we have made the following explanations and modifications.

The explanatory variables in this study are "going to the local personal clinic, community hospital and general hospital for medical treatment when ill = 1, no = 0", "going to the local cold clinic = 1, no = 0", "having received the follow-up assessment, health examination and other services for some diseases provided by the local community health service center (station)/township health center free of charge = 1, no = 0" as the proxy variables of medical service utilization. Although they do not fully represent the utilization of medical services, they can partly explain the choice of migrant workers’ medical service utilization. From the perspective of economics, China's migrant population has made great contributions to industrialization and urbanization. In the context of Healthy China Initiative, promoting the timely medical treatment for them in case of illness, and encouraging the initiative of using medical services are the inherent requirements. However, due to the paradox of "economic acceptance - social exclusion", the migrant population in a strange environment could not make full use of medical services with insufficient psychological integration, low health literacy and underdeveloped social network, and choose to go to hospital when ill, and make regular physical examination, etc. Therefore, this paper starts with the public health education of migrant workers, hoping that immigrants can receive more public health education, and improves their psychological integration, health literacy and social network to increase the medical service utilization efficiency.

This paper further explains them in the text (in blue). As this study is about migrant population with labor capacity from the perspective of public health education, to explore the influencing factors of medical service utilization of migrant population, the study object and content are relatively clear, not to involve the relevant ethical issues. We thank for the tips from the reviewer, which provides a reference for further clarifying the study object and scope of the article, and we also marked it in the text.

Comment 2):

The details provided about the characteristics of the study population are still insufficient, considering that the scope is to compare this population with other potential populations in different geographical areas. Some necessary details are age distribution (median and quartiles), ethnic information, and timing information about the migration moment, i.e. years from the migration or whether they are first or second generation migrants.

Answer: Thank you very much for the suggestions on specifying characteristics of the research population. We absolutely agree on it. Therefore, three key characteristics of the migrant population, namely age, migrant year and migrant distance, are added into the study. The specific description is as follows.

According to the question of whether the respondents were born before or after 1980, we divide the migrant workers into two groups: "the new generation" (<40 years old) and "the old generation" (≥ 40 years old). Moreover, those who have moved for more than 5 years are classified as "long-term migrants", and those who have moved for less than or equal to 5 years are the "short-term migrants". The migrant workers who flow across provinces are grouped as "trans provincial migration", and the others belong to the "non trans provincial migration". In order to simplify the analysis, we classify the group who participate one or more health education as "accepted" and no health education as "not accepted". The variables of medical service utilization are set as "seeking medical treatment" and "not seeking medical treatment". It can be seen from the cross-analysis table that, no matter the new generation or the old generation, long-term immigrants or short-term immigrants, trans provincial migration or non-trans-provincial migration groups, after receiving public health education, the probability of choosing to use medical services has expanded, which to some extent indicates that migrant workers' receiving public health education can promote their behavior of using medical services. After receiving health education, the proportion of the new generation and the old generation of migrant workers who choose to see a doctor is equal, but the behavior of long-term migrants to use medical services is nearly 2 percentage points higher than that of short-term migrants, and the use of medical services for the trans province group is also nearly 2 percentage higher than the non-trans, which indicates that the longer the migrant workers' migration years, the better the public health education will be for the use of medical services, so is the migration distance.

Table. Cross analysis on migration characteristics of migrant workers with the use of medical service after public health education

New

generation

Old

generation

Long-term

immigrant

Short-term

 immigrant

Trans

province

Non trans

province

Not accept

Accept

Not accept

Accept

Not accept

Accept

Not accept

Accept

Not accept

Accept

Not accept

Accept

Seeking medical

53.30%

47.05%

53.75%

47.96%

52.53%

46.06%

53.75%

47.96%

53.37%

46.92%

52.85%

47.40%

Not seeking medical

46.70%

52.95%

46.25%

52.04%

47.47%

53.94%

46.25%

52.04%

46.63%

53.08%

47.15%

52.60%

Total

100%

100%

100%

100%

100%

100%

100%

100%

100%

100%

100%

100%

Many thanks & Best regards

Author Team

Reviewer 2 Report

Thank you for taking into account the comments, I believe you now present a stronger paper. 

Author Response

Dear Reviewer,

We thank you for your great support in the first round review again. It really helps us to improve this study. 

We understand your comments and we've already revised the manuscript as you saw. May some of the issues be difficult to be overcome in this article due to the limitations, we will continuously promote it in the further study.

Best regards

Author team